# Deficient Incorporation of Rabies Virus Glycoprotein into Virions Enhances Virus-Induced Immune Evasion and Viral Pathogenicity

**DOI:** 10.3390/v11030218

**Published:** 2019-03-04

**Authors:** Chunfu Li, Hongliang Zhang, Lina Ji, Xiao Wang, Yongjun Wen, Guangpeng Li, Zhen F. Fu, Yang Yang

**Affiliations:** 1The State Key Laboratory of Reproductive Regulation and Breeding of Grassland Livestock, School of Life Sciences, Inner Mongolia University, Hohhot 010070, China; lcf52082@163.com (C.L.); nn930817@163.com (L.J.); wangxiao@life.imu.edu.cn (X.W.); imudwzx@163.com (G.L.); 2College of Veterinary Medicine, Inner Mongolia Agricultural University, Hohhot 010018, China; zhanghongliang001@126.com (H.Z.); yongjunwen@126.com (Y.W.); 3Department of Pathology, College of Veterinary Medicine, University of Georgia, Athens, GA 30602, USA; zhenfu@uga.edu; 4The State Key Laboratory of Agricultural Microbiology, College of Veterinary Medicine, Huazhong Agricultural University, Wuhan 430070, China

**Keywords:** rabies virus, immune evasion, glycoprotein incorporation, laboratory-attenuated rabies virus, wild-type rabies virus, dendritic cell activation, membrane fusion

## Abstract

Previous studies have shown that wild-type (wt) rabies virus (RABV) evades the host immune response by restricting expression of glycoprotein (G), which blocks activation of dendritic cells (DCs) and induces production of virus-neutralizing antibodies (VNAs). In the present study, wt RABVs not only restricted G expression but also reduced incorporation of G into mature virions compared with laboratory-adapted viruses. A recombinant RABV expressing triple G was used to further determine whether G expression relates to incorporation. The recombinant virus showed higher expression and incorporation of G and activated more DCs than the virus that expressed a single copy of G. Removal of G from viruses using subtilisin or Dithiothreitol (DTT)/ Nonidet P-40 (NP40) almost completely abolishes DC activation and VNA production. Consequently, these G-depleted viruses cause lethal infection in mice. Thus, wt RABVs can subvert DC-induced antiviral immune response and maintain pathogenicity by decreasing G expression in infected cells and G incorporation into virions.

## 1. Introduction

RABV, a single-stranded and negative-sense RNA virus in the *Rhabdoviridae* family, causes more than 59,000 human fatalities annually throughout the world [1,2]. RABV consists of five structural proteins, the nucleoprotein (N), phosphoprotein (P), matrix protein (M), glycoprotein (G), and RNA-dependent RNA polymerase (RdRp, also termed large protein, L). G is the only protein of RABV that is glycosylated and exposed on the surface of the virion [3]. G binds cellular receptors and facilitates cellular entry via direct fusion with the cellular membrane [4]. Moreover, G is the only protein that can induce virus-neutralizing antibodies (VNAs) [5].

RABV G plays an essential role in viral immune evasion [6,7,8,9,10,11,12]. The wt RABV fails to induce VNAs in human patients [13] and in experimentally infected animals, such as mice [14], skunks [15], ferrets [16], and dogs [8]. Low expression of G is the major reason that wt RABVs escape early detection by the innate immune system of the host [11]. In contrast to wt RABVs, most laboratory-adapted RABVs replicate quickly and express high levels of G, thereby inducing strong immune responses and high levels of VNAs in animals [7,9]. Reverse genetics has resulted in construction of several recombinant rabies vaccines with more than one copy of G expressed. Introduction of additional copies of G significantly improves immunogenicity and stimulates increased production of VNAs than in the parental virus [17,18,19]. A recombinant RABV expressing two copies of G incorporated 1.5-fold higher levels of G into RABV, suggesting that G incorporation relates to the level of G expression [18]. In addition, a G-deficient RABV was no longer able to spread through the host cell [20]. Thus, the level of G incorporation into viral particles may affect the ability of the virus to activate the immune system and infect cells.

DCs are the most efficient antigen-presenting cells. Recombinant RABVs expressing innate immune genes stimulated the production of higher levels of VNAs and provided better protection by activating more DCs than the parental virus [12,14,21,22]. By contrast, wt RABV does not efficiently induce DC activation [6,23]. The wt RABV or laboratory-adapted RABV expressing G from the wt virus bind DCs less efficiently, produce lower levels of leader RNA in DCs, and consequently induce lower levels of VNAs than those of laboratory-adapted viruses [6]. However, G expression levels affect incorporation into virions and consequently immune evasion. The observation that pathogenic strains of RABV express lower levels of G is well established in the literature, as is the key role G plays in immune evasion. However, how wt RABV uses G to evade immune recognition remains poorly understood.

In the present study, we investigated the mechanism(s) by which reduced G expression affects G incorporation into virions and immune evasion. We found that wt RABVs expressed lower levels of G and incorporated less G into virions compared with laboratory-adapted RABV. A recombinant RABV carrying three copies of G expressed higher levels of G and incorporated more G into virions compared with the parental virus, enhancing G-dependent cell fusion and DC activation. In addition, removal of G from virions blocked DC activation and enhanced pathogenicity. These findings support a novel mechanism, in which wt RABV evades DC recognition by restricting G incorporation into mature virions.

## 2. Materials and Methods

### 2.1. Mice

C57BL/6 mice were purchased from Vital River Laboratories (China) and bred under specific pathogen-free conditions in the Research Center for Laboratory Animal Science at Inner Mongolia University (Inner Mongolia permit number: 2014-0002).

### 2.2. Ethics Statement

All animal experiments were performed according to the National Institutes of Health Guide for the Care and Use of Laboratory Animals [24]. Experimental protocols were approved by the Institutional Animal Care and Use Committee at Inner Mongolia University.

### 2.3. Cells and Virus

Mouse neuroblastoma (NA) cells were maintained in RPMI 1640 media (Gibco, Suzhou, China) supplemented with 10% fetal bovine serum (FBS) (Gibco). BSR cells, a cloned cell line derived from BHK-21 cells, were maintained in Dulbecco’s modified Eagle’s medium (DMEM; Gibco) containing 10% FBS. Myeloid DCs were generated as previously described [25]. Briefly, bone marrow was removed from tibias and femur bones of BALB/c mice. Following red blood cell lysis and washing, progenitor cells were plated in RPMI 1640 media supplemented with 10% FBS, 0.1 mM nonessential amino acids, 1 mM sodium pyruvate, and 20 ng/mL recombinant murine granulocyte-macrophage colony-stimulating factor (Peprotech, Rocky Hill, United States) in 6-well plates at 4 × 10^6^ cells/well. DCs display low levels of CD86, which is characteristic of immature DCs. B2c is a laboratory-attenuated RABV generated from CVS-24 by serial passaging in BHK-21 cells [11]. CVS is the standard challenge virus. DRV-Mexico (DRV) is a wt RABV isolated from a rabid dog in Mexico in the 1990s [26]. Additionally, B2c (DRV-P) and B2c (DRV-G) are two recombinant viruses, in which the B2c P and G proteins, respectively, were replaced with those from DRV [7,27]. Viral stocks were prepared as described previously [9]. Briefly, one-day-old suckling mice were inoculated with 10 μL viral inoculum by intracerebral (i.c.) route. Moribund mice were euthanized, and brains were removed. A 10% (*w*/*v*) suspension was prepared by homogenizing the brain in DMEM. The homogenate was centrifuged to remove debris, and the supernatant was collected and stored at −80 °C. For BPL inactivation, BPL (Sigma) was diluted at a final concentration of 1:4000 (*v*/*v*) and incubated for 24 h at 4 °C to ensure complete viral inactivation. Subsequently, BPL was inactivated for 1 h at 37 °C.

### 2.4. Construction of RABV Expressing Triple G

The B2c vector pcDNA/B2c and N-, P-, G-, and L-expressing helper plasmids were generated as previously described [7,19,28]. RABV G1, G2, and G3 gene sequences of triple G recombinant RABV [B2c-3G] were amplified from the full-length pcDNA/B2c plasmid using the following specific primers: G1 forward (5′-AGACTTAATTAAAGATGGTTCCTCAGG-3′ [Pac I site underlined]) and G1 reverse (5′-CACGTACGCGGCCTTCACAGTCTGATCTCAC-3′ [BsiW I site underlined]), G2 forward (5′-AAGGCCGCGTACGTGGTTCCTCAGGTTC-3′ [BsiW I site underlined]) and G2 reverse (5′-TCGCTAGCCTTCACAGTCTGATCT-3′[Bmt I site underlined]), and G3 forward (5′-AAGGCTAGCGAAAGATGGTTCCTCAGGTT-3′ [Bmt I site underlined]) and G3 reverse (5′-CTCTGGATGAGAAGTGTTGCTAGCTGTT-3′ [Nhe I site underlined]). PCR products were digested with enzyme sets Pac I & BsiW I, BsiW I & Bmt I, and Bmt I & Nhe I and were ligated into full-length pcDNA/B2c. Recombinant cDNA clones B2c-3G were constructed by switching the coding sequence of the original G with triple copies of the same G from B2c without altering the start/stop signals outside the ORF of each G. Each G has its own open reading frame (Figure 3A).

### 2.5. Western Blot Analysis

Western blot analysis was performed as described previously [9]. Briefly, NA cell extract or purified virions were resolved by 10% sodium dodecyl sulfate polyacrylamide gel electrophoresis (SDS-PAGE) and transferred onto a polyvinylidene difluoride (PVDF) membrane. The membrane was blocked with TBS containing 1% bovine serum albumin (BSA) and incubated with respective primary antibodies overnight at 37 °C followed by horseradish peroxidase-conjugated secondary antibodies (Sigma) for 1 h at room temperature. Proteins were detected using west pico chemiluminescent substrate (Thermo Fisher Scientific, United States). Band signals corresponding to immunoreactive proteins were measured and scanned for integrated optical density (IOD) using Image-Pro Plus 6 (Media Cybernetics, United Kingdom). Anti-RABV-G (G53) or N (N42) were prepared as previously described [29].

### 2.6. Viral Titration

Viral titration experiments were performed with FITC-labeled anti-rabies virus N protein antibodies (Fujirebio Diagnostics, Malvern, United States) in RABV-infected NA cells as described previously [30]. NA cells (1 × 10^5^ cells/well in a 96-well plate) were inoculated with serial 10-fold dilutions of virus and incubated at 34 °C for 48 h. Cells were then fixed with 80% ice-cold acetone for 30 min. Cells were washed twice with PBS and then stained with FITC-conjugated anti-RABV N antibodies at 37 °C for 1 h. Antigen-positive foci were counted under a fluorescence microscope (Zeiss, Jena, Germany), and the viral titer was calculated as fluorescent focus units per milliliter (FFU/mL). All titrations were carried out as quadruplicates.

### 2.7. Virus-Neutralizing Antibody Tests

Sera were collected from mice to measure VNAs using fluorescent antibody virus-neutralization tests as previously described [6]. Briefly, 50 μL of serial 5-fold dilutions of sera were prepared in 96-well microplates in 100-μL volumes. CVS-11 (50 μL) was added to each chamber and incubated at 37 °C for 90 min. NA cells (5 × 10^5^ cells/mL) were added to each chamber, and the plates were incubated at 37 °C for 24 h. The plates were then fixed with ice-cold 80% acetone and stained with FITC-conjugated anti-RABV N antibodies. Twenty fields in each chamber were observed under a fluorescence microscope, and the 50% endpoint titers were calculated according to the Reed–Muench formula [31]. The values were compared to those obtained with reference sera (National Institute for Biological Standards and Control, Herts, United Kingdom) and normalized to international units (IU)/mL.

### 2.8. Virion Purification, Electron Microscopy, and Immunogold-Labeling

To quantify G incorporation into RABV virions, viruses were purified by sucrose-density gradient ultracentrifugation, either alone or followed by filtration with 3000 MWCO Amicon Ultra-15 Centrifugal Filter Units (EMD Millipore, Tullagreen, Ireland). The level of G in RABV virions was calculated by immunogold-labeling transmission electron microscopy (TEM) as described previously [32]. In brief, viruses were adsorbed onto parlodion-coated nickel grids for 30 min followed by fixation with 2.5% glutaraldehyde for 30 min. Grids were then floated on a drop of tris-buffered saline (TBS), pH 7.4, for 5 min, followed by floating on drops of TBS containing 1% BSA for 1 h with RABV-G-specific monoclonal antibody (G53) diluted 1:300 in TBS containing 1% BSA. After washing with TBS three times, samples were incubated for 1 h with goat anti-mouse IgG coupled to 5-nm gold particles diluted 1:10 in TBS containing 1% BSA. Grids were again washed with TBS and then stained with 2% phosphotungstic acid (pH 7.0) for 30 s. The grids were then examined using a JEOL 1230 transmission electron microscope (JEOL, Tokyo, Japan). Incorporation of G proteins was quantified by calculating the number of gold particles on the surface of virions as previously described [33].

### 2.9. Protease and Salt Treatment of Virions

To remove G from the exterior of RABVs, 100 µg purified RABVs were treated with 1 mL NP40 (50 mM Tris–HCl pH 8.0, 10 mM DTT, 0.05% NP40) for 20 min at 4 °C. To stop the reaction, 200 µg glycogen in 1 mL ddH_2_O were added for 30 min as described [34,35]. The treated virus (2 mL) was concentrated to 30 µL using 100K MWCO Amicon Ultra-2ml Centrifugal Filter Units. To prevent destruction of the virion lipid structure by DTT/NP40, the outer layer of the viral membrane was removed by subtilisin (SUB) digestion as previously described [33,36]. In brief, concentrated virus (equivalent to 50 µg of protein) was incubated with 100 µg SUB (Sigma, United States) in 20 mM Tris-Cl (pH 8) and 1 mM CaCl_2_ for 30 min at 37 °C. To neutralize SUB, the virus was diluted to 1 mL with TNE buffer (10 mM Tris–HCl, pH 8.0, 100 mM NaCl, and 2 mM Na_2_-EDTA) containing 5 µg phenylmethanesulfonyl fluoride (PMSF, Sigma) for 15 min at room temperature. The virus was concentrated using 3000 MWCO Amicon Ultra-15 Centrifugal Filter Units. All samples were subjected to western blot analysis or electron microscopy.

### 2.10. Purification of RABV by Ultracentrifugation

B2c was grown in NA cells at a multiplicity of infection (MOI) of 0.01 at 34 °C. After four days, a 1-liter volume of supernatant was collected and centrifuged at 3000× g for 5 min to remove cellular debris. The clarified supernatant was then layered onto serial dilutions of 35, 45, 55, and 65% sucrose-density gradients and centrifuged at 210,000× *g* for 3 h in an Optima L-80 XP ultracentrifuge (Beckman Coulter, Brea, USA). The band was collected, diluted in PBS, pelleted (210,000× *g* for 1 h), and precipitated with 20% trichloroacetic acid (TCA). The viral pellet was separated by 8–12% SDS-PAGE, and the RABV-G- and N-specific bands were verified by western blot using anti-RABV-G (G53) or anti-RABV N (N42) antibodies.

### 2.11. Quantitative Real-Time PCR

Total viral RNA was extracted from the supernatant with Viral RNA Mini Kit (Qiagen, Valencia, USA) and used for qRT-PCR in an Applied Biosystems^®^ 7500 Real-Time PCR System (Thermo Fisher Scientific, Singapore) as described previously [6]. Each reaction was carried out in triplicate with approximately 100 ng DNase-treated RNA and 5 nM of each primer pair using Brilliant II SYBR green qRT-PCR master mix kit (Stratagene, Cedar Creek, USA) according to the manufacturer’s instructions. For absolute quantification of viral genomic RNA (vRNA), complementary DNA (cDNA) were synthesized by SuperScript III Reverse Transcriptase (Invitrogen, Carlsbad, USA) following the manufacturer’s instructions using RABV-specific primers (v_1250R: AGTCCTCGTCATC AGAGTTGAC). Furthermore, qPCR was performed using primer v_1121F: GGAAAAGGGACATTTGAAAGAA and v_1250R. A standard curve was generated from serially diluted RNA in vitro transcribed from plasmids expressing RABV N, and the copy numbers of viral RNA were normalized to 1 mg total RNA [22].

### 2.12. Enzyme-Linked Immunosorbent Assay (ELISA)

To analyze antibody responses, sera were obtained from mice immunized with non-treated, DTT/NP40-, or SUB-treated RABV. Mouse anti-RABV-G IgG ELISA kits (Alpha Diagnostic, San Antonio, USA) were used to detect RABV-specific antibodies. B2c and B2c-3G were titrated in NA cells and diluted to 10^5^, 10^4^, and 10^3^ FFU/mL. To evaluate the content of RABV-G, a RABV-G ELISA kit from Wuhan Institute of Biological Products (China) was used. All assays were performed according to the manufacturer’s instructions, and color development was measured by Flash Multimode Reader (Thermo Fisher Scientific).

### 2.13. Low pH-Induced G-Dependent Cell Fusion Assay

Cell fusion assays were performed as previously described with minor modifications [37,38]. BSRs, NAs, or DCs were grown to 80% confluence and then infected with different RABVs at an indicated MOI. After 2 h at 37 °C, the supernatant was discarded, and fresh culture media were added. Cells were incubated at 37 °C for another two days or directly used for the next step. Cells were treated with fusion buffer (10 mM Na_2_HPO_4_-10 mM NaH_2_PO_4_-150 mM NaCl-10mM 2-(N-morpholino) ethanesulfonic acid) at pH 5.3 and incubated for 2 min at room temperature. After removing the fusion buffer, cells were incubated at 37 °C for 1 h. Cells were fixed with 80% ice-cold acetone and stained with FITC-conjugated anti-RABV N antibodies or fuchsine solution. Calculation of the cell fusion index was performed. The ratio of number of multinucleated cells to number of total nuclei was determined from photographs of 10 randomly selected fields, in which 300 nuclei or more were counted [38].

### 2.14. Viral Endocytosis Assay

For the viral internalization assay, cells were incubated with RABVs at 4 °C for 2 h and washed three times with PBS, followed by culturing at 37 °C. After 4 h, cells were treated with trypsin without EDTA (10 min for DCs, 4 min for NAs) to remove surface-bound but not internalized virus and then washed with PBS three times. Total RNA was extracted from the NA and DC cells using TRIzol (Life Technologies, Inc., Carlsbad, USA). The amount of internalized viral RNA was quantified by viral qRT-PCR.

### 2.15. Statistical Analysis

Statistical significance of differences between groups was determined using Student’s *t*-test with *** indicating *p* value < 0.0001, ** *p* value < 0.001, and * *p* value < 0.05 using Graph Pad Prism software.

## 3. Results

### 3.1. G Expression in Cells and G Incorporation into Virions of the wt RABV Are Less Than Those Observed in Laboratory-Attenuated RABV

Our recent studies demonstrated that wt RABV evades DC-mediated immune activation by expressing low levels of G in infected cells, which results in inefficient binding/entry into DCs [6]. To determine if G expression in infected cells affects G incorporation into virions, NA cells were infected with a laboratory-attenuated RABV (B2c), a wt RABV (DRV), a recombinant B2c expressing DRV G [B2c (DRV-G)], or a recombinant B2c expressing DRV-P [B2c (DRV-P)] at MOI of 1. To exclude the possibility that differences in the level of G expression are due to differences in the binding affinity of the detecting antibodies, an in vitro translation system was employed to synthesize G from CVS-B2c and DRV as described previously [10]. To examine the binding affinity of antibodies to G, the synthesized proteins were immunoprecipitated with G53 and then detected by western blot using streptavidin-HRP. As shown in Appendix A, similar amounts of B2c G and DRV-G were detected (IOD 166 vs. 177). As shown in Figure 1A, a significantly higher level of G was detected in cells infected with CVS-B2c or B2c (DRV-P) than in those infected with DRV or B2c (DRV-G). By contrast, the levels of N and P proteins were similar in cells infected with each of the viruses. The ratio of band signals corresponding to the G and N immune-reactive proteins (G/N ratio) was higher in cells infected with B2c (0.97) or B2c (DRV-P) (1.04) than that in cells infected with DRV (0.59) or B2c (DRV-G) (0.20) (Figure 1A). These data confirm that wt RABV selectively restricts expression of G [7,9,10,11,39].

To investigate whether the level of G expression affects incorporation into virions, RABV was purified from the four-day post-infection (dpi) supernatants of cells infected with each of the viruses at MOI 0.1. After sucrose gradient ultracentrifugation, there were two bands of virus materials, from 45% to 55% and 55% to 65%. The smaller band may contain defective interfering (DI) particles; thus, only the larger band was used to perform SDS-PAGE. The virion G/N ratios of B2c (1.03), DRV (0.39), B2c (DRV-P) (1.17), and B2c (DRV-G) (0.33) were similar to those found in infected cells (Figure 1B). To further eliminate contamination, virions purified by ultracentrifugation were concentrated using 3000 MWCO Amicon Ultra-15 Centrifugal Filter Units for electron microscopy. The concentrated virus was adsorbed onto parlodion-coated nickel grids and stained with the anti-G monoclonal antibody G53. As shown in Figure 1C, typical bullet-shaped virions were found in CVS-B2c and B2c (DRV-P). However, virions of DRV and B2c (DRV-G) were rounder and shorter than those of B2c and B2c (DRV-P) but maintained the typical shape of rhabdoviruses. The envelope of DRV or B2c (DRV-G) appeared thinner than that of B2c or B2c (DRV-P). Immune electron microscopy of virions revealed a significantly higher average number of gold particles per virion on the surface of virions from CVS-B2c (17.2) or B2c (DRV-P) (23.3) than on those from DRV (2.6) or B2c (DRV-G) (4) (Figure 1D). Moreover, fewer than two gold particles were detected in 71% of the labeled DRV and 35% of B2c (DRV-G) virions, indicating that RABVs expressing G from wt viruses incorporate lower levels of G in budding virions than those expressing G from laboratory-adapted RABVs. Taken together, these results suggest that wt RABV incorporates fewer G molecules into mature virions compared with laboratory-adapted RABVs. The incorporation of G protein in budding virions largely depends upon the levels of G expression in infected cells.

### 3.2. Enhancement of G Expression Increases Its Incorporation into Virions

In contrast to wt RABVs, most attenuated RABVs express high levels of G and induce strong adaptive immune responses that protect the host from rabies infection. To examine whether overexpression of G further enhances G incorporation into virions, a recombinant RABV, B2c-3G, which contains three copies of G, was constructed from the B2c full-length plasmid [7] (Figure 2A). To ensure triple expression of G, each copy of G has its own open reading frame as described previously [17,19]. The replication rates of the parental B2c and B2c-3G were determined in NA and BSR cells by single-step growth kinetics. As shown in Figure 2B, the replication rate of B2c was similar to that of B2c-3G during the first two days in both NA and BSR cells. By 3 dpi (BSR cells) or 4 dpi (NA cells), the viral titer of B2c-3G was significantly higher than that of B2c.

To test whether the additional G gene altered viral morphology, virions of B2c-3G were observed by TEM. The average diameters of B2c and B2c-3G virions were 85 μm and 105 μm, respectively. Moreover, the average length of B2c and B2c-3G virions were 180 μm and 360 μm, respectively. These observations indicated that the average diameter and length of B2c-3G were significantly longer than those of the parental virus (Figure 2C). Similarly, previous studies reported that enhancement of the VSV or RABV viral genome extends the length of virions [18,40].

At 3 dpi, to avoid contamination of cellular vesicles, the supernatant was purified by ultracentrifugation and subjected to ELISA to evaluate G incorporation into the virions (Figure 2D). At 10^5^ FFU/mL, the OD450 value of G incorporated into virions was 2.5 times higher in cells infected with B2c-3G than that of cells infected with B2c (*p* < 0.05). Moreover, the OD450 at B2c-3G viral titer 10^3^ FFU/mL was still significantly higher than that of 10^5^ FFU/mL B2c. To further confirm that increasing G expression in cells resulted in higher G incorporation into virions, the G/N ratios of RABV-infected NA cells and sucrose gradient-purified virions were determined by western blot analysis (Figure 2E). The G/N ratios were 1.59 and 0.86 in cells infected with B2c-3G and B2c, respectively, and the G/N ratios were 1.16 and 0.71 in purified B2c-3G and B2c virions, respectively. Thus, G incorporation into B2c-3G was much higher than G incorporation into B2c in the purified virions. Thus, these results further confirm that increasing G expression in infected cells increases incorporation of G into virions.

### 3.3. Removal of G from RABV Virion by Subtilisin Digestion or DTT/NP40 Treatment

Previous studies reported that a G-deficient RABV lost the ability to infect cells. It is challenging to produce sufficient amounts of G-deficient RABV by reverse genetics [20]. To further determine that G is absolutely required for the RABV replication cycle and immune evasion, the outer layer of RABV was removed by two methods that have previously been used to remove integral membrane proteins from virions [33,34]. One approach was to incubate virions with DTT/NP40, which has been shown to efficiently separate major membrane proteins and virion lipids of vaccinia virus from the virion surface [34]. After treatment with DTT/NP40, the lipid was removed, and viral infectivity diminished [35]. The RABV structure was analyzed after treatment by TEM after negative staining with phosphotungstic acid. A different form of the virion was observed (Figure 3A). After treatment with SUB or DTT/NP40, the virion surfaces became discontinuous and contained fractures, which may be due to partial removal of the virion protein. However, the virion size and bullet shape after treatment with either agent remained similar to that of non-treated virions. These results indicate that treatment with either agent can remove the outer protein from the virion surface without changing the number and basic shape of the viral particles. To provide additional evidence that G anchored on the viral membrane was almost completely removed by the treatment, immunogold-labeling of ultra-15 centrifugal filter units was performed on purified B2c. Virions were stained with anti-RABV-G antibodies. Following negative staining, only a few or no gold particles were located on the outer layer of the B2c virion. An average of 1.8 gold particles per virion (total 11 RABV particles) was detected on SUB-treated B2c virions, and an average of 1.9 gold particles per virion (total 12 virus particles) was detected on DTT/NP40-treated B2c under observation by electron microscopy (Figure 3B). Compared with B2c staining with G antibody-labeled gold particles in Figure 1C, there were significantly fewer gold particles on treated B2c than on non-treated B2c. The same amounts of B2c and B2c-3G were loaded to perform the DTT/NP40 or SUB treatment, and the G/N ratios were determined by western blot analysis. The G/N ratios for B2c, DTT/NP40-treated B2c, and SUB-treated B2c were 0.48, 0.29, and 0.14, respectively. The G/N ratios for B2c-3G, DTT/NP40-treated B2c-3G, and SUB-treated B2c-3G were 0.92, 0.70, and 0.11, respectively (Figure 3C). These results indicate that G can be removed from RABV surfaces by SUB or DTT/NP40 treatment. SUB treatment was more efficient than DTT/NP40 treatment in removing G from virions.

After treatment with DTT/NP40 or SUB, RABVs were concentrated by Amicon Ultra-15 Centrifugal Filter to determine the gene copy numbers by qRT-PCR with specific primers. As shown in Figure 3D, there were no significant differences in gene copy numbers between treated and non-treated viruses. Effects of removing G from the viral membrane on viral infection and production were determined in NA cells with a single-step growth curve. Comparison of time courses of viral replication in NA cells infected with B2c or B2c-3G revealed that the replication rates of SUB- and DTT/NP40-treated RABV were severely decreased (more than 2 logs) before 4 dpi, indicating that removal of G from virions blocks early infection (Figure 3E). However, the viral titers for SUB- or DTT/NP40-treated RABVs increased rapidly by 5 dpi and were similar to that of non-treated virus. These results suggest that SUB and DTT/NP40 remove G from the virion surface without influencing the basic structure and genomic RNA of the virus.

### 3.4. Removal of G Incorporation Reduces Viral-Inducing Cell Fusion

Previous studies found that RABV enters the cell through the endocytic pathway, and the viral envelope fuses with the endosomal membrane after acidification [41]. Because G is the unique outer component of RABV, a low pH-dependent cell fusion assay was performed as described previously [38] to determine whether the level of G expression and/or incorporation affects cell-to-cell fusion. This assay simulates virus-induced endosomal membrane fusion, which is the uncoating step of the RABV replication cycle [4]. As shown in Figure 4A, BSR and NA were infected with 1 MOI of B2c, B2c-3G, DRV, B2c (DRV-P), or B2c (DRV-G) and the N protein of RABV was stained with FITC-labeled antibody. At 2 dpi, all infected cells showed similar levels of N protein expression at pH 7.2. However, when the pH of media was changed to 5.3, more than 80% of cells infected with B2c or B2c (DRV-P) showed cell membrane fusion. As in uninfected cells, no cell fusion was observed in cells infected with DRV or B2c (DRV-G). B2c-3G induced the strongest cell membrane fusion among all of the viruses (Figure 4B). To further confirm that the level of G incorporation affects viral uncoating by influencing cell membrane fusion, the same gene copies number (10^12^ copies/mL) of sucrose gradient-purified RABVs were incubated with BSR cells for 4 h and then treated with fusion media (pH 5.3) for 0.5 h. As shown in Figure 4C, B2c-3G induced the strongest cell fusion among all the viruses, whereas DRV did not induce any cell fusion. Cell fusion was inhibited when B2c-3G and B2c were treated with DTT/NP40 or SUB. However, BPL treatment did not affect viral induction of cellular membrane fusion. Thus, removal of G from virions restricted G-dependent cell membrane fusion, providing evidence that wt RABV does not efficiently infect DCs due to its restriction of G-induced cell membrane fusion.

### 3.5. The Level of G Incorporation Affects Viral Internalization and DC Activation

To determine whether the level of G incorporation alters viral endocytosis in vitro, B2c, DRV, B2c-3G, B2c (DRV-G), B2c (DRV-P), or B2c and B2c-3G treated with SUB or DTT/NP40 were used to infect NA cells. The viral titer and viral genomic RNA copy numbers were determined after 4 dpi. As shown in Figure 5A, the ratio of viral RNA to viral titer was 10^3^ for DRV and 10^2.8^ for B2c (DRV-G), which is 1-log higher than those of B2c (10^1.8^) and B2c (DRV-P) (10^1.9^) and 2-log higher than B2c-3G (10^0.9^). A significantly higher viral RNA/viral titer ratio was detected when the virus was treated with SUB or DTT/NP40 (10^5.5^). These data revealed a reverse correlation between the level of G incorporation and the viral RNA/viral titer ratio, indicating that higher G incorporation leads to increased viral replication in NA cells. DCs were infected with B2c-3G, G-depleted RABV, B2c, DRV, B2c (DRV-G), or B2c (DRV-P) at RNA copy numbers of 10 copies/cell to determine the infectivity of each RABV in DCs. Viral RNA was analyzed by qRT-PCR 4 h post-infection (hpi). As shown in Figure 5B, significantly less viral RNA was detected in DCs infected with RABV after G was removed by SUB or DTT/NP40.

To further examine DC activation induced by RABVs with different levels of G incorporation, expression levels of CD80, CD86, and MHCII were determined using flow cytometry. DCs were incubated with 1, 10, and 100 gene copies/cell. As shown in Figure 5C, the levels of CD86, CD80, and MHCII expression increased in a gene copy number-dependent manner in DCs infected with CVS-B2c, B2c (DRV-P), or B2c-3G but not in DCs infected with DRV, B2c (DRV-G), or RABVs treated with SUB or DTT/NP40. Furthermore, RABVs entirely lost the ability to activate DCs when the viral genomic RNA was inactivated by UV or BPL. The levels of these costimulatory molecules on DCs infected with DRV, B2c (DRV-G), or RABV treated with SUB or DTT/NP40 did not change after increasing virus doses, demonstrating that the inability of wt RABVs or G-deficient RABVs to activate DCs is not dependent on the infection dose.

### 3.6. Pathogenicity of RABV Is Inversely Correlated with the Level of G Incorporation into Virions

To determine if the level of G incorporation affects RABV pathogenicity, RABVs treated with or without SUB or DTT/NP40 were used to infect mice. Suckling C57BL/6 mice were infected with 10^6^ gene copies of RABV by intracranial (i.c.) injection. As shown in Figure 6A, 100% of the infected mice required euthanasia before 9 dpi and had similar viral titers (10^6^ FFU/mL) in the brains. Although the onset of disease and death were delayed for several days in mice inoculated with SUB- or DTT/NP40-treated RABV compared with those infected with non-treated B2c or B2c-3G, all infected mice eventually succumbed to rabies. These data indicate that all of the RABVs treated with or without SUB or DTT/NP40 can induce lethal infection when directly inoculated into the CNS.

To determine if the level of G incorporation mediates peripheral RABV immune evasion, C57BL/6 mice were infected intramuscularly (IM) with RABV at 10^6^ gene copies/mouse and observed daily for clinical signs for 21 days (Figure 6B,C). All adult mice inoculated with B2c-3G, B2c, or treatment buffer only (containing 10 mM DTT/0.05% NP40 or 100 µg/mL SUB) through IM injection did not develop disease regardless of mouse age and weight. However, for mice with body weights of 12 to 14 g, the mortality rates of mice infected with DTT/NP40- or SUB-treated B2c or B2c-3G ranged from 20% to 40% (Figure 6B). For mice with body weights of 18 to 20 g, only one mouse infected with SUB-treated B2c-3G developed clinical signs of rabies (Figure 6C). These results indicated that G-deficient RABVs had significantly increased pathogenicity and presented a higher risk for young mice than for adult mice.

### 3.7. Immune Evasion Is Inversely Correlated with the Level of G Incorporation into Virions

To examine whether decreased G incorporation further reduces immunogenicity, suckling mice were infected with RABVs before or after treatment with SUB or DTT/NP40. In mice that succumbed to rabies, the average viral titer reached 10^4.3^ FFU/mL in B2c-infected mice and 10^3^ FFU/mL in B2c-3G-infected mice (Figure 7A). In mice infected with B2c-3G or B2c, an average VNA production of 1.38 IU/mL or 1.14 IU/mL was detected, respectively (Figure 7B). However, in mice infected with RABVs treated with SUB or DTT/NP40, the average VNA level was less than 0.5 IU/mL similar to that of the mock-infected group. Analysis of the G-specific antibody using an anti-RABV-G ELISA kit revealed that the anti-G antibody level in mice infected with SUB- or DTT/NP40-treated RABVs was 500 U/mL similar to that of mock-infected mice. By contrast, the level of anti-G antibody reached 750 U/mL in mice infected with B2c or B2c-3G (Figure 7C). To further assess the immune responses mediated by RABV with different levels of G incorporation, activation of DCs, T cells, and B cells were analyzed in the spleens of mice at 3 and 9 dpi. As shown in Figure 7D, significantly more activated DCs (CD11c^high^ and CD86^high^), T helper 17 (Th17^high^) cells (CD4^high^ and IL-17^high^), and plasma cells (CD138^high^) were detected in the spleens of mice infected with non-treated B2c or B2c-3G when compared with those of mice infected with DRV or mock-infected mice. By contrast, there were no significant differences in the numbers of activated DCs, Th17 cells, and plasma cells in the spleens of mice infected with SUB- or DTT/NP40-treated RABVs when compared with those of mice infected with DRV or mock-infected mice. These data demonstrate that wt RABVs or G-deficient RABVs fail to induce immune responses, such as activation of T cells and production of VNAs.

## 4. Discussion

The RABV-G protein is widely recognized as an important determinant for rabies pathogenesis and immune responses [6,7,9,10,11,12]. The wt RABV evades the host immune responses by expressing low levels of G and failing to activate DCs [6,9]. In the present study, the low level of G expression in cells infected with wt RABV resulted in low G incorporation into virions. Thus, the low level of G incorporation leads to failure of wt RABV to activate DCs, because RABV G is the only surface protein of the virion that induces endosome fusion, promoting its internalization [41]. Indeed, recombinant RABV expressing three copies of G incorporated more of G into virions and consequently activated more DCs. By contrast, removal of G from RABV virions by proteases almost completely abolished DC activation.

Rabid human patients do not develop VNAs at the time of death [13]. Similarly, laboratory animals experimentally infected with wt RABV do not develop VNAs [8,14]. Thus, wt RABVs have developed various ways to evade host immune system recognition. Many RABV proteins have been implicated in this immune evasion. For example, RABV P protein inhibits IRF3/7, STAT1/2/3, and IKKε phosphorylation [17,18,19]. The RABV N protein evades activation of the RIG-I-mediated antiviral response [42]. The RABV M protein interacts with RelAp43, a member of the NF-κB family, to block interferon (IFN) secretion [43]. The earliest host-cell response to viral infection is recognition of pathogen-associated molecular patterns (PAMPs) followed by type-I IFN system activation. RABV can induce IFN through an IPS-1-dependent pathway in DCs [44]. However, wt RABV can only induce an incomplete viral replication cycle, and hardly any viral proteins were expressed in DCs after RABV infection [6,44]. Thus, the failure of DC activation after RABV infection is dependent on G but independent of P, N, and M protein-related pathways.

Our previous studies demonstrated that the inability of wt RABV to stimulate the production of VNAs is due to the failure of wt RABV to activate DCs [6,8]. Adoptive transfer of DCs primed with wt RABV ex vivo did not result in activation of DCs or production of VNAs in the recipients [6]. As a consequence, these recipients were not protected against challenge with virulent RABV. This is in contrast to laboratory-adapted RABV, which expresses high levels of G [9,10,11], activates DCs in vitro and in vivo [6,8,22,45], and stimulates high production of VNAs [6,9,12,22,45]. This is most likely due to the fact that wt RABV restricts G expression [9,10,11]. In the present study, the low level of G expression resulted in a low level of G incorporation into virions. In fact, wt RABV incorporates 10 times fewer G molecules into virions than laboratory-adapted viruses. Although it is established that wt RABV restricts G expression [9,11], the present study indicates that it is the low level of G incorporation that leads to the failure of wt RABV to activate DCs. Increasing G expression by expressing three copies of G (B2c-3G) increased G incorporation into virions and increased DC activation. By contrast, removal of G from laboratory-adapted B2c or B2c-3G abolished the ability to activate DCs. RABV-G is the only surface protein of RABV that assists viral binding and uncoating [20]. The DC-mediated immune evasion by wt RABV is due to its inefficient binding and subsequently low level of leader RNA transcription in DCs [6]. Our analysis indicates that the low level of G expression and incorporation lead to minimal cell-to-cell fusion, resulting in low RABV infection and little to no DC activation. This was further confirmed by the removal of G from laboratory-adapted virions, which almost completely abolished their ability to activate DCs.

To exclude the possibility that removal of the lipid structure by DTT/NP40 destroys the viral genome, a SUB digestion procedure was performed. Previous studies showed that SUB treatment of HIV-1 and influenza virus removes proteins outside the virion without destroying the viral genome [33,36]. Indeed, in our studies, removal of G from RABV by SUB did not abolish infectivity in NA cells or animals. In NA and BSR cells, the replication rate of DTT/NP40 or SUB-treated RABVs was significantly lower than that of non-treated RABVs during the first 4 dpi but similar by 5 dpi. Pathogenicity analyses of suckling mice clearly demonstrated that RABVs treated with SUB or DTT/NP40 replicated in the CNS and induced rabies with clinical signs occurring 1–2 days later than with non-treated virus. However, the pathogenicity of G-deficient B2c or B2c-3G increased compared with that of untreated B2c or B2c-3G in young mice infected intramuscularly. This strongly suggests that decreasing G incorporation levels increases RABV pathogenicity, most likely due to the ability of these viruses to evade immune stimulation.

In summary, our results indicate that the level of G incorporation into viral particles is an important mechanism by which wt RABV avoids DC activation, resulting in immune evasion. The level of G expression in infected cells determines the level of G incorporation, which in turn determines DC activation. Thus, the level of G incorporation plays a key role in the evasion of DC recognition and activation.

## Figures and Tables

**Figure 1 viruses-11-00218-f001:**
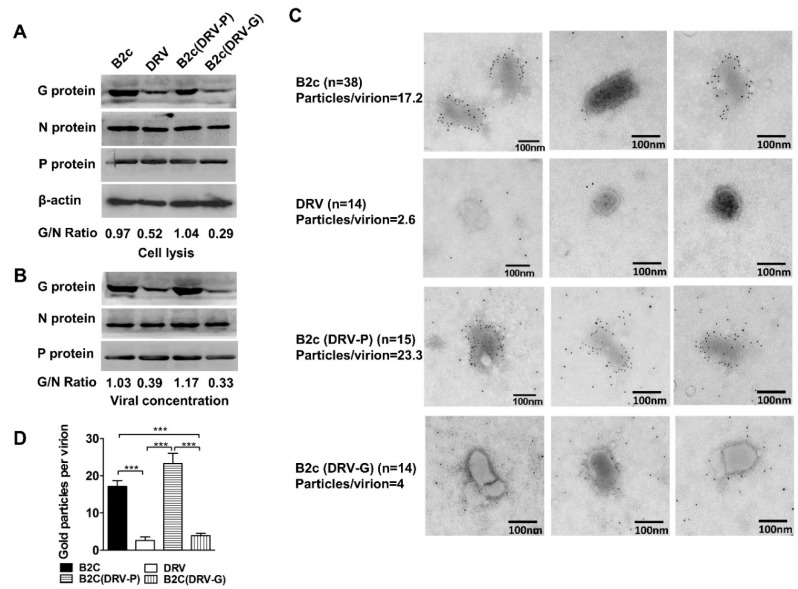
Expression of RABV-G in NA cells and G incorporation into virions. (**A**) Western blot of viral protein expression in NA cells infected with indicated RABVs at MOI 1. (**B**) Western blot analysis of viral protein incorporated into virions. The ratio between G and N was calculated from the intensity of the band using Image-Pro Plus 6.0. (**C**) Electron micrographs of the indicated RABV virions purified from the supernatants of infected NA cells. RABV-G protein was detected by staining with RABV-G-specific antibodies labeled with 5-nm immunogold particles. (**D**) The number of gold particles per virion was calculated by manual counting of positive gold dots. Significance of differences between the gold particles per virion of the indicated virus was assessed by the unpaired *t*-test. *, *p* < 0.05; **, *p* < 0.01; ***, *p* < 0.001.

**Figure 2 viruses-11-00218-f002:**
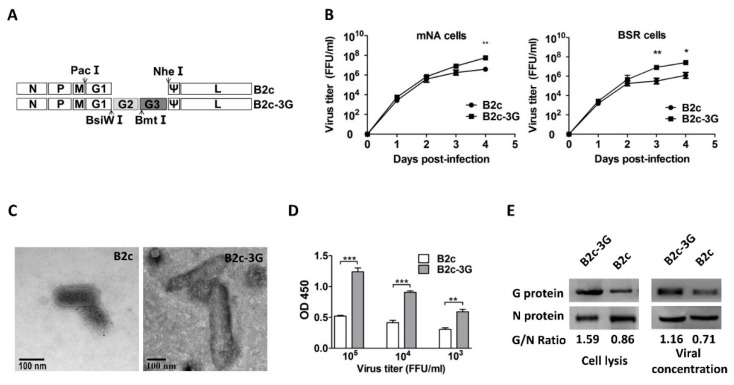
Construction and in vitro characterization of B2c-3G, a recombinant RABV expressing three copies of G. (**A**) Schematic diagram of B2c recombinant virus carrying two extra G genes, each with its own reading frame. The arrows indicate the location of restriction enzyme sites in the genome of RABV. (**B**) Growth curves of B2c-3G and B2c were generated in NA and BSR cells. (**C**) Morphological analysis and length of B2c-3G and B2c were compared by electron microscopy after negative staining. (**D**) ELISA of G content in RABVs at 10^3^, 10^4^, and 10^5^ FFU/ml. (**E**) Western blot analysis of viral protein in NA cells and viral protein incorporated into virions. The ratio between G and N was calculated from the intensity of the band using Image-Pro Plus 6.0. The significance of differences between viral titers or OD450 of B2c-3G and B2c were assessed by the unpaired *t*-test. *, *p* < 0.05; **, *p* < 0.01; ***, *p* < 0.001.

**Figure 3 viruses-11-00218-f003:**
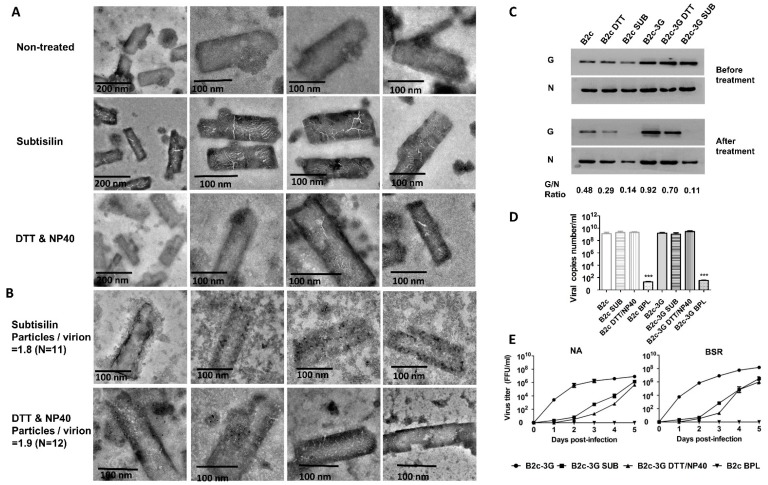
Removal of G from RABV virions after treatment with DTT/NP40 or subtilisin. (**A**) Morphological analysis of DTT/NP40- or subtilisin-treated RABV by electron microscopy after negative staining. (**B**) RABV-G protein on virions was detected by staining with RABV-G-specific antibodies labeled with 5-nm immunogold particles. (**C**) Western blot analysis of viral N and G proteins incorporated into virions. (**D**) Viral genomic RNA from RABV treated with or without DTT/NP40 or subtilisin and analyzed by qRT-PCR. (**E**) Growth curves of B2c-3G and B2c in the NA and BSR cells.

**Figure 4 viruses-11-00218-f004:**
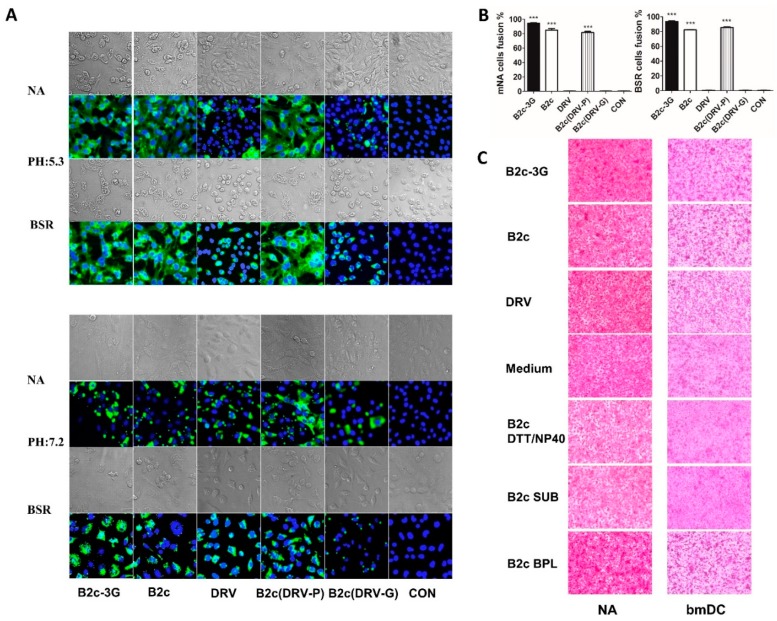
RABV-G-mediated cell-to-cell fusion. (**A**) BSR and NA cells were infected with the indicated viruses for 48 h followed by microscopic evaluation of cell-to-cell fusion at pH 7.2 or pH 5.3. (**B**) Cell fusion index was determined by calculating the ratio of the number of nuclei of multinucleated cells to the number of total nuclei in the photographs taken of randomly selected fields, in which 300 nuclei or more were counted. (**C**) NA and bmDC cells were treated with the indicated viruses for 4 h before microscopic evaluation of cell-to-cell fusion at pH 5.3.

**Figure 5 viruses-11-00218-f005:**
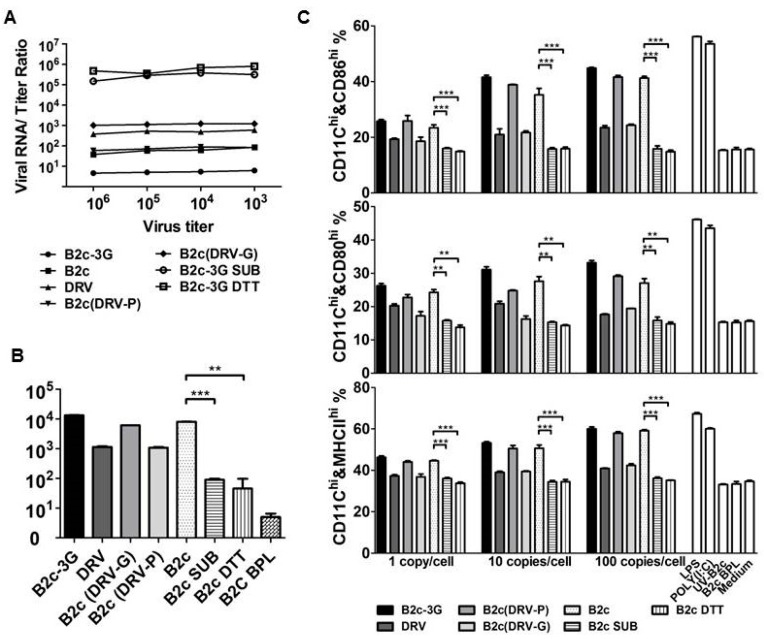
DC activation by different RABVs in vitro. (**A**) RABV genomic RNA/titer ratio of the indicated viruses detected by qRT-PCR and viral titration, respectively. (**B**) RABV-binding efficiency evaluated by qRT-PCR. (**C**) Flow cytometric analysis of the number of CD80-, CD86-, and MHC- positive DCs infected with CVS-B2c or DRV treated with 100 µg of subtilisin protease or medium control buffer. Significance of differences between the percentage in normal and infected DCs were assessed by the unpaired *t*-test. *, *p* < 0.05; **, *p* < 0.01; ***, *p* < 0.001.

**Figure 6 viruses-11-00218-f006:**
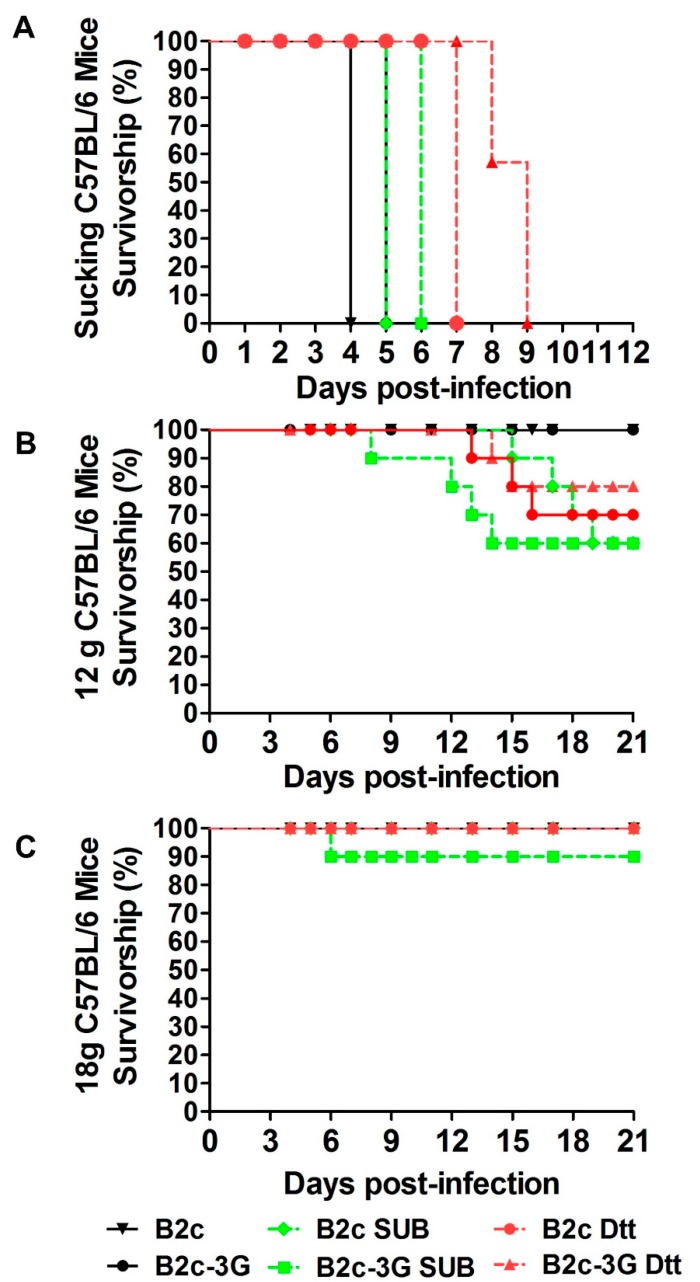
Removal of G from RABV virions increased viral pathogenicity. (**A**) Survivorship of suckling mice infected with the indicated viruses or through i.c. inoculation. The statistical significance of differences in survival rates were analyzed by Kaplan–Meier plots (*n* = 10 in each group, log rank *p* < 0.05). (**B**) Survivorship of 12 g mice infected with the indicated viruses through IM inoculation. (**C**) Survivorship of 18 g mice infected with the indicated viruses through IM inoculation.

**Figure 7 viruses-11-00218-f007:**
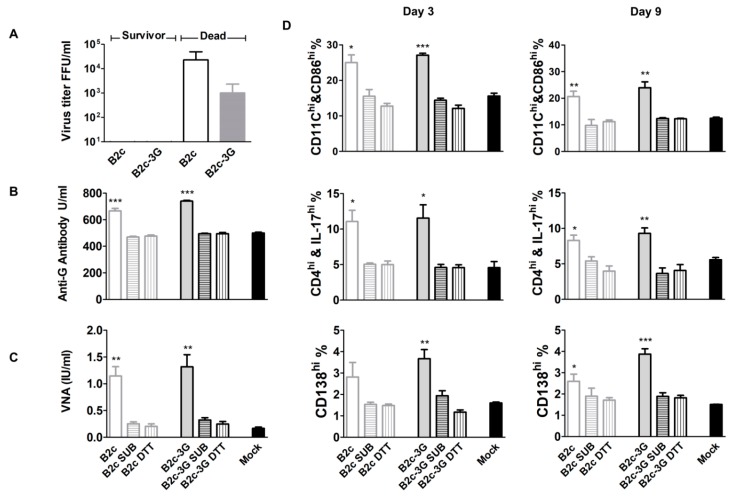
Removal of G from RABV virions abolished immune activation. (**A**) Viral titers in the brains of dead and surviving mice. (**B**) VNAs in mice infected with different viruses. (**C**) ELISA of anti-RABV-G antibodies in mice infected with different viruses. (**D**) FACS analysis of the activation status of DCs, T cells, and B cells in the spleens of mice infected with different viruses at 3 or 9 dpi. The significance of differences between the positive cell percentages in mock and infected mice were assessed by the unpaired *t*-test. *, *p* < 0.05; **, *p* < 0.01; ***, *p* < 0.001.

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
