# Peer review of "Deficient Incorporation of Rabies Virus Glycoprotein into Virions Enhances Virus-Induced Immune Evasion and Viral Pathogenicity"

_viruses, 2019, doi:10.3390/v11030218_

Round 1

Reviewer 1 Report

General comments

This paper reinforces earlier findings made by the authors of this paper and other investigators indicating that anti-viral immunogenicity and pathogenicity is mainly determined by the amount of RABV G expressed on the virus envelope or on the surface cell membrane of RABV-infected cells. While the results obtained with live wt RABVs, attenuated RABVs, or recombinant RABVs convincingly demonstrate that restricted G expression is associated with a decrease in anti-RABV immune responses and an increase in viral pathogenicity, I have problems with the experiments in which G-deleted virions were used. For example, there are inconsistencies in the treatment of virions. The rationale why in some experiments a combination of NP40 and DTT and in others only DTT was used for the removal of G  is not clear. In addition, the conclusions drawn from data obtained with detergent- or subtilisin-treated virions are somewhat doubtful. Since  it is well known that the G is essential for viral infection (particularly for attachment and penetration), the removal of G will consequently result in a reduction of infectivity which in turn will account for most if not all the observations made with G-deleted virions including decreased fusion activity, decrease in  DC activation, and increased pathogenicity. Therefore, PBL-inactivated virions should be included as controls in all experiments performed with G-depleted virions (except determination of pathogenicity in mice).

Specific comments

1.      The micrographs shown in Figures 3A and B are of poor quality and it is difficult to see differences between treated and non-treated samples.

2.      The Western blot analysis shown in Fig 3C shows only very little effect of DTT treatment on the amount of G. Furthermore, very little N is seen in the subtilisin-treated samples as compared to the other samples suggesting that much less subtilisin-treated virions were loaded.  The results would be more convincing if similar amounts of each sample were loaded.

3.      Legend for Figure 4D is missing.

Author Response

Response to Reviewer 1 Comments

General comments

Point 1: The rationale why in some experiments a combination of NP40 and DTT and in others only DTT was used for the removal of G is not clear.

Response 1: The virions were treated with DTT/NP40 not DTT. We apologize for this error. We have corrected the methods in our revised manuscript as suggested.

Point 2: In addition, the conclusions drawn from data obtained with detergent- or subtilisin-treated virions are somewhat doubtful. Since it is well known that the G is essential for viral infection (particularly for attachment and penetration), the removal of G will consequently result in a reduction of infectivity which in turn will account for most if not all the observations made with G-deleted virions including decreased fusion activity, decrease in DC activation, and increased pathogenicity. Therefore, PBL-inactivated virions should be included as controls in all experiments performed with G-depleted virions (except determination of pathogenicity in mice).

Response 2: We agree with the reviewer. We have included a PBL-inactivated virus as a control. As shown in Figures 3D and 3E, PBL treatment almost completely destroyed the viral genomic RNA and blocked viral replication. PBL-treated RABV retained the ability to promote low pH-induced cellular membrane fusion. However, the virus could not activate DCs, because the viral genomic RNA was destroyed (Figure 4C, Figure 5B, and Figure 5C).

Specific comments

1.     The micrographs shown in Figures 3A and B are of poor quality and it is difficult to see differences between treated and non-treated samples.

Response 1: We thank the reviewer for this comment. As suggested, some of the EM pictures in Figures 3A and 3B have been replaced with higher quality images. Although there were not substantial differences between the subtilisin-treated and DTT/NP40-treated virions, treated virions appeared different from non-treated virions. There were significant cracks on the viral surface after treatment. We revised the manuscripts as shown below.

Line 345: “A different form of the virion was observed (Figure 3A). After treatment with SUB or DTT/NP40, the virion surfaces became discontinuous and contained fractures, which may be due to partial removal of the virion protein. However, the virion size and bullet shape after treatment with either agent remained similar to that of non-treated virions.”

The main purpose of this experiment was to confirm that treatment with subtilisin or DTT/NP40 did not destroy the basic structure of the rabies virus. However, the main difference between the treated and non-treated groups was the increased number of anti-G-labeled immunogold particles on the virion membrane after subtilisin or DTT/NP40 treatment.

2.     The Western blot analysis shown in Fig 3C shows only very little effect of DTT treatment on the amount of G. Furthermore, very little N is seen in the subtilisin-treated samples as compared to the other samples suggesting that much less subtilisin-treated virions were loaded. The results would be more convincing if similar amounts of each sample were loaded.

In order to determine the loading amount of virions before treatment, a western blot was performed with untreated virus before subtilisin or DTT/NP40 treatment. As shown in Figure 3C, the same amounts of N and G were loaded. We repeated the subtilisin and DTT/NP40 experiment and loaded equivalent amounts of each sample. Both subtilisin and DTT/NP40 treatment reduced G incorporation on the RABV virion, Subtilisin caused greater removal of G from the virion as compared with DTT/NP40 treatment. These results were further supported by our mice experiments, which showed that the subtilisin-treated virus was more pathogenic than the DTT/NP40-treated virus. Subtilisin treatment also decreased the level of N protein on the virion. Subtilisin can sightly digest N on the virion without destroying the viral genomic RNA.

We have revised the manuscript as shown below.

Line 366: “The same amounts of B2c and B2c-3G were loaded to perform the DTT/NP40 or SUB treatment, and the G/N ratios were determined by Western blotting. The G/N ratios for B2c, DTT/NP40-treated B2c, and SUB-treated B2c were 0.48, 0.29, and 0.14, respectively. The G/N ratios for B2c-3G, DTT/NP40-treated B2c-3G, and SUB-treated B2c-3G were 0.92, 0.70, and 0.11, respectively (Figure 3C). These results indicate that G can be removed from RABV surfaces by SUB or DTT/NP40 treatment. SUB treatment was more efficient than DTT/NP40 treatment in removing G from virions.”

3.     Legend for Figure 4D is missing.

Response 3: We have corrected this error. The previous Figure 4A and Figure 4B were incorporated into Figure 4A in the revised manuscript.

Reviewer 2 Report

Li et al. present a potentially interesting paper about the connection between the expression levels of the RABV glycoprotein and DC cell activation state.  The working hypothesis is that the overexpression of RABV G leads to DC activation when using laboratory adapted viruses, conversely, street viruses have less G on the virion, thus evade DC activation.  The experiments raise several interesting questions for future work, namely, since DCs are not a target for RABV, and are the middle man in terms of immunological process, how does what is presented in the manuscript play out in a pathogenicity model for a host that is naturally susceptible to rabies?

The details of the experiments and results were difficult to follow due to grammar.  The English must be improved before further consideration of the manuscript.

Author Response

Response to Reviewer 2 Comments

Point 1:

Li et al. present a potentially interesting paper about the connection between the expression levels of the RABV glycoprotein and DC cell activation state. The working hypothesis is that the overexpression of RABV G leads to DC activation when using laboratory adapted viruses, conversely, street viruses have less G on the virion, thus evade DC activation. The experiments raise several interesting questions for future work, namely, since DCs are not a target for RABV, and are the middle man in terms of immunological process, how does what is presented in the manuscript play out in a pathogenicity model for a host that is naturally susceptible to rabies?

Response: We agree with the reviewer’s comments. In our experiments, the younger mice were susceptible to RABV infection, which has also been demonstrated in previous studies (1, 3). T cells are activated by the RABV vaccine, leading to B cell activation and subsequent production of rabies virus-neutralizing antibodies, which offers host protection. Young hosts with compromised immune systems or immune-deficient hosts may have increased susceptibility to RABV pathogenicity. Our current findings demonstrate that loss of G protein expression and incorporation lead to immune escape and infection. We recently found that RABV G degradation may occur naturally by digestive enzymes in the host mouth and ubiquitination of the RABV glycoprotein, reducing viral glycoprotein incorporation and enhancing RABV pathogenicity. Furthermore, these G incorporation deficient virion were unable to fully activate dendritic cells. Although, DCs are the middle man of immunological process, some previouse studies have proved that the B cells deficient and IPS-1 pathway knockout mice were susceptible to infection of RABVs (2, 4). Thus, the Rabies virus fewer packaging glycoproteins would be more effective to avoid the recognition of dendritic cells and other immune cells to effectively invade the central nervous system. In future experiments, we will further examine the relationship between G incorporation and dendritic cells in a CD11c-deficient mouse model.

Point 2:

The details of the experiments and results were difficult to follow due to grammar. The English must be improved before further consideration of the manuscript.

We have rewritten the whole text and gammar to increase conciseness. Corrections were made for subject-verb agreement, comma placement, consistent use of hyphens and abbreviations, and run-on sentences.

This manuscript has been proofread and edited by a professional English editing company, BioScience Writers (BSW), located in Houston, Texas. A certificate of editing from BSW can be provided upon request.

1.          Faber, M., J. Li, R. B. Kean, D. C. Hooper, K. R. Alugupalli, and B. Dietzschold. 2009. Effective preexposure and postexposure prophylaxis of rabies with a highly attenuated recombinant rabies virus. Proceedings of the National Academy of Sciences of the United States of America 106:11300-11305.

2.          Faul, E. J., C. N. Wanjalla, M. S. Suthar, M. Gale, C. Wirblich, and M. J. Schnell. 2010. Rabies virus infection induces type I interferon production in an IPS-1 dependent manner while dendritic cell activation relies on IFNAR signaling. PLoS Pathog 6:e1001016.

3.          Guo, C., C. Wang, S. Luo, S. Zhu, H. Li, Y. Liu, L. Zhou, P. Zhang, X. Zhang, Y. Ding, W. Huang, K. Wu, Y. Zhang, W. Rong, and H. Tian. 2014. The adaptation of a CTN-1 rabies virus strain to high-titered growth in chick embryo cells for vaccine development. Virology journal 11:85.

4.          Hooper, D. C., T. W. Phares, M. J. Fabis, and A. Roy. 2009. The production of antibody by invading B cells is required for the clearance of rabies virus from the central nervous system. PLoS Negl Trop Dis 3:e535.

Round 2

Reviewer 1 Report

The authors responded appropriately to my comments, The changes made in the manuscript are satisfactory and the quality of the paper has been significantly improved.  

Reviewer 2 Report

Li et al present a heavily modified and highly improved manuscript describing the impact of G expression from wild type and recombinant viruses.  The data demonstrate that overexpression of G increases infectivity.  Lower G expression may lead to reduced immune evasion suggesting that one of the differences in pathogenicity between laboratory and street viruses is mediated by G expression.  Lab viruses expresses higher level of G which results in a higher level of immune activation.  The experiments are sufficiently designed and could be further supported by in-vivo analysis, but such experiments are beyond the scope of the current manuscript.